# The relationship between conference presentations and in-hospital mortality in patients admitted with acute myocardial infarction: A retrospective analysis using a Japanese administrative database

**Daisuke Takada[1], Yuki Kataoka[2], Tetsuji Morishita[1], Noriko Sasaki[1], Susumu Kunisawa[1], Yuichi Imanaka[1,3] ***

1 Department of Healthcare Economics and Quality Management, Graduate School of Medicine, Kyoto University, Kyoto, Japan, 2 Department of Internal Medicine, Kyoto Min-Iren Asukai Hospital, Kyoto, Japan, 3 Department of Health Security System, Centre for Health Security, Graduate School of Medicine, Kyoto University, Kyoto, Japan

* imanaka-y@umin.net

**Data Availability Statement:** Basically, the Japanese government has made a law "Act on the

## Abstract

### Background

Clinicians' research activities reportedly improve their healthcare performance. Presenting research at conferences may be related to improved patient care outcomes; however, few studies have investigated this relationship. Thus, we examined the association between presenting at conferences and the mortality of patients hospitalized for acute myocardial infarction.

### Methods

We analyzed an administrative database of acute care hospitals in Japan. The study compared patients admitted to hospitals in which physicians made at least one conference presentation during the patient's admission year (Presentation Group) with those admitted to hospitals with no conference presentations (Control group). We performed multivariable logistic regression analyses to estimate the risk of all-cause in-hospital mortality. Five models were fitted: a Crude model, unadjusted; Model 1, adjusted for personal factors, including sex, age, Killip classification, and so on; Model 2, adjusted for Model 1 plus hospital factors; Model 3 was a multilevel analysis clustered by hospital codes and adjusted for the same variables as Model 1; Model 4 was adjusted for Model 1 plus evidence-based practices through causal mediation analysis.

### Results

After excluding 3,544 patients with missing Killip classification or ambulance use, data for 56,923 patients in 384 acute care hospitals were analyzed. Drug prescription in accordance with the evidence was significantly greater in the Presentation group than in the Control

Protection of Personal Information", which restricts the use of personal information. Our dataset includes potentially sensitive information, such as diagnosis and medical histories regarded as "Special care-required personal information" specified by the above law. And if our dataset would be combined with other datasets, it can be linked to personal identification. Moreover, our datasets were collected from each hospital within the bounds of the ethical procedures of academic research and the data use were limited to the researchers listed on the research plan submitted in advance to the Ethics Committee. So our data are not permitted to release publicly. The Japanese ethical guideline "Ethical Guidelines for Medical and Health Research Involving Human Subjects" has imposed these restrictions in consistency with the law "Act on the Protection of Personal Information". Requests to access the data should be submitted to Kyoto University Graduate School of Medicine/Faculty General Affairs Division, Research Promotion Section. Email: 060kensui@mail2.adm.kyoto-u.ac.jp.

**Funding:** This study was funded by Ministry of Health, Labour and WelfareHealth and Labour Sciences Research Grants/Grant (JPMH21IA1005, JPMH22AA2003) and Japan Society for the Promotion of Science (JP23H00448). The funders had no role in study design, data collection and analysis, decision to publish, or preparation of the manuscript.

**Competing interests:** The authors have declared that no competing interests exist.

group. Moreover, conference presentation was significantly associated with lower in-hospital mortality in all models (Odds ratios (OR) = 0.68, 95% Confidence intervals (CIs): 0.65 to 0.72 in the Crude model; OR = 0.73, 95% CIs: 0.68 to 0.79 in Model 1; OR = 0.76, 95% CIs: 0.70 to 0.82 in Model 2; OR = 0.84, 95% CIs: 0.76 to 0.92 in Model 3; OR = 1.00, 95% CIs: 0.92 to 1.09 in Model 4).

## Conclusion

The promotion of scholarly activities such as conference presentations might improve patient outcomes through increased evidence-based practice.

## Introduction

Improving health care performance is a critical goal for healthcare systems worldwide [1]. Healthcare organizations may improve their performance through clinicians engaging in research activities with research institutions. This assumption has led to policies that promote clinicians' research engagement to improve health care performance [2–5].

There are several mechanisms of performance improvement at individual and institutional levels. Conducting research activities can develop research networks [2] and enhance clinicians' absorptive capacity—healthcare providers' willingness to implement the evidence-based practices generated from their research [5]. Research activities can also include participating in trials; doctors in trial-participating hospitals change their prescription practices quicker, and in-hospital mortality in these hospitals is lower in some departments due to the "trial effect" [4]. Incidentally, the beneficial effects of clinicians' research activities are reported in cases of myocardial infarction [6–8], an area in which many randomized clinical trials are conducted [9].

Various research activities have the potential to improve health care performance, but the effectiveness of other research activities remains unexplored. Scholarly pursuits, a type of research activity, are mandatory for residents in many countries [10, 11], and presenting at conferences is the first stage of scholarly pursuits in Japan. Presenting research at conferences may promote the characteristics in clinicians that lead to improved health care performance at individual and institutional levels. However, there are few studies specifically investigating the relationship between conference presentations given by hospital physicians and the outcomes of patient care at an institutional level.

Our hypothesis is that clinicians' conference presentations might be associated with improved patient outcomes through increased evidence-based practice. Therefore, we examined the association between clinicians' conference presentations and mortality in acute myocardial infarction (AMI) cases after adjusting for potentially confounding patient characteristics. We further confirmed that the proportions of prescribed medical therapies were optimal according to treatment guidelines for evidence-based practice.

## Materials and methods

### Study population

We retrospectively analyzed an administrative database of Japanese acute care hospitals participating in the Quality Indicator/Improvement Project (QIP). In brief, the Japanese government has adopted a public health care payment system, the Diagnosis Procedure Combination (DPC)/Per-Diem Payment System [12], which is currently used by more than 80% of the acute

care hospitals in Japan. More than 500 of these hospitals joined QIP [13]. The DPC data comprises a Form 1, an EF-file, and other files. Form 1 contains important clinical summaries, including date of birth, sex, admission dates, discharge outcome, diagnosis, surgical procedure, and various severity scores such as the Killip classification upon admission and other clinical information. The International Classification of Diseases, 10th Revision (ICD-10) codes were used for diagnosis in the DPC database. The EF-file contains information on medical resource utilization during hospitalization, including medications administered and other medical procedures. Nearly all the hospitals participating in the QIP are educational institutions that hire board-certified cardiologists. We accessed this database for this research from September 13, 2019 for research purpose, and we did not access the information that could identify individual participants during or after data collection.

The following two inclusion criteria were used: (i) inpatients with a record of acute myocardial infarction (AMI) (I21.x in the 2013 version of the ICD-10) in both the trigger and principal diagnoses between April 1, 2014 and December 31, 2018; (ii) those aged 18 years or over [14]. And we excluded patients with missing covariates.

## Exposure

A "conference presentation" was defined as a presentation at the annual scientific meeting of the Japanese Circulation Society (JCS), which is one of Japan's most attended meetings for cardiologists; the JCS also certifies cardiologists in Japan. We divided the study population into two groups based on whether doctors at the admitting hospital had made conference presentations during the year that the patient was admitted (Presentation group and Control group). When we confirmed a presentation, we investigated the number of presentations for each hospital on the web page of the University Hospital Medical Information Network [15].

## Outcome and statistical analyses

The primary outcome measure was all-cause in-hospital mortality. First, we prepared multivariable logistic regression models and a multilevel logistic regression model to estimate the risk of mortality, after adjusting for confounding factors, and confirmed the goodness of fit using c-statistics. To show accordance with evidence-based practice, we also included the following prescribed drugs during hospitalization in both groups: aspirin, P2Y12 Receptor Inhibitors, beta blocker, angiotensin converting enzyme (ACE) inhibitor or/and Angiotensin II Receptor Blocker (ARB) and statin [16].

We fitted five models: a Crude model was unadjusted; Model 1 was adjusted for sex, age, the Killip classification, smoking, ambulance use, hypertension, atrial fibrillation, old myocardial infarction, diabetes, renal disease and chronic obstructive pulmonary disease (personal factors); Model 2 was adjusted for Model 1 plus admitted year and the interquartile range of the number of admissions at each hospital per year (hospital factors). Model 3 was a multilevel analysis clustered by hospital codes and adjusted for the same variables as Model 1. Model 4 was adjusted for Model 1 plus the evidence-based practices through causal mediation analysis. All the covariates were detected on admission.

Sensitivity analyses were basically performed to adjust for the same covariates in Model 2. These analyses involved 1) excluding university hospitals and their affiliated hospitals, 2) restricting patients to those who stayed more than two days, and 3) restricting presentations to those in which the clinician was the first author only, 4) additionally adjusting for the Barthel index when the data did not contain any missing measurements, 5) excluding patients who were readmitted during the observation period.

In all of the logistic regression analyses, we produced the odds ratios (ORs) and 95% confidence intervals (CIs) for the variables. A two-sided significance level of 0.05 was used, data cleaning was performed using Microsoft SQL server 2014 and all analyses were conducted using R version 3.4.1 (The R Development Core Team, Vienna, Austria).

## Ethical consideration and data availability

The Ethics Committee, Graduate School of Medicine, Kyoto University, approved this study (approval number: R0135). This study was conducted in accordance with the Ethical Guidelines for Medical and Health Research Involving Human Subjects of the Ministry of Health, Labour and Welfare, Japan. These guidelines require the protection of patient anonymity. The data were anonymized, and the requirement for informed consent was waived.

## Results

### Characteristics of study population

We identified a total of 60,567 patients in 384 acute care hospitals admitted due to acute myocardial infarction. The flow chart for patient selection is illustrated in Fig 1. We excluded patients with missing Killip classification (n = 3,474) and ambulance use (n = 70) data. Thus, 56,923 patients were ultimately included in the complete case analysis (Fig 1).

### Patient characteristics and prescriptions of optimal medical therapies

Patient characteristics are shown in Table 1. As indicated, in-hospital mortality was 7.7% in the Presentation group and 10.9% in the Control group. The number of cases in which drugs were prescribed after admission was significantly greater in the Presentation group than in the Control group for all drugs considered in the study (p-value: < .001).

### Multivariable logistic regression analysis for in-hospital mortality

Table 2 shows the results of the multivariable analysis of in-hospital mortality after adjusting for the covariates in the three models. As indicated, clinicians' conference presentations were significantly associated with lower in-hospital mortality in all models, except Model 4 (OR = 0.68, 95% CIs: 0.65 to 0.72 in the Crude model; OR = 0.73, 95% CIs: 0.68 to 0.79 in Model 1; OR = 0.76, 95% CIs: 0.70 to 0.82 in Model 2; OR = 0.84, 95% CIs: 0.76 to 0.92 in Model 3; OR = 1.00, 95% CIs: 0.92 to 1.09 in Model 4).

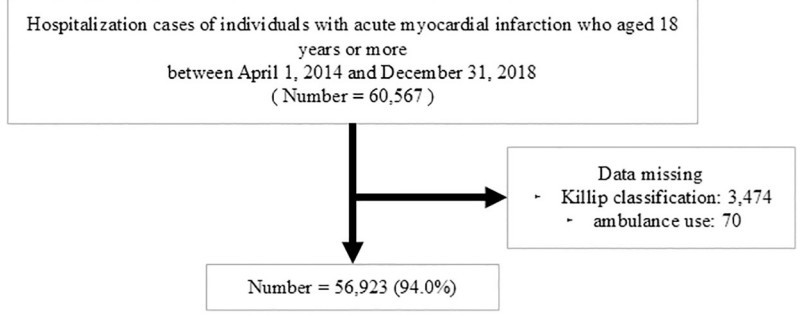

**Fig 1. Patient flow chart.**

**Table 1. Demographic and clinical characteristics of the populations.**

| | Control group[a] | Presentation group[a] | P value |
|---|---|---|---|
| **Total number of patients** | 22531 | 34392 | |
| **Number of presentations at the year (mean)** | none | 4.25 | <0.001 |
| **Sex: female (%)** | 6170 (27.4) | 9008 (26.2) | 0.002 |
| **Age (mean (SD))** | 69.93 (13.41) | 69.65 (13.04) | 0.01 |
| **Killip classification (%)** | | | <0.001 |
| 1: no clinical signs of heart failure | 10178 (45.2) | 17352 (50.5) | |
| 2: rales or crackles in the lungs, and elevated jugular venous pressure | 6515 (28.9) | 9182 (26.7) | |
| 3: frank acute pulmonary edema | 2035 (9.0) | 3037 (8.8) | |
| 4: cardiogenic shock or hypotension, and peripheral vasoconstriction | 3803 (16.9) | 4821 (14.0) | |
| **Ambulance use (%)** | 14510 (64.4) | 24370 (70.9) | <0.001 |
| **Smoking** | 10686 (47.4) | 16716 (48.6) | 0.006 |
| **Comorbidities** | | | |
| Hypertension (%) | 14828 (65.8) | 23718 (69.0) | <0.001 |
| Atrial fibrillation (%) | 2024 (9.0) | 3268 (9.5) | 0.037 |
| Old myocardial infarction (%) | 489 (2.2) | 731 (2.1) | 0.712 |
| Diabetes (%) | 6372 (28.3) | 9970 (29.0) | 0.075 |
| Renal disease (%) | 1282 (5.7) | 2087 (6.1) | 0.065 |
| Chronic Obstructive Pulmonary Disease (%) | 796 (3.5) | 1222 (3.6) | 0.917 |
| **Prescribed drugs after admission** | | | |
| Aspirin (%) | 20623 (91.5) | 32586 (94.7) | <0.001 |
| P2Y12 Receptor Inhibitors (%) | 19656 (87.2) | 31113 (90.5) | <0.001 |
| β-blocker (%) | 13993 (62.1) | 23858 (69.4) | <0.001 |
| Angiotensin-converting-enzyme-inhibitor or angiotensin receptor blocker (%) | 14326 (63.6) | 24989 (72.7) | <0.001 |
| Statin (%) | 17365 (77.1) | 28982 (84.3) | <0.001 |
| **Revascularization therapy after admission** | | | |
| Percutaneous coronary intervention (%) | 19279 (85.6) | 30982 (90.1) | <0.001 |
| Coronary artery bypass grafting (%) | 333 (1.5) | 756 (2.2) | <0.001 |
| **In-hospital mortality (%)** | 2457 (10.9) | 2658 (7.7) | <0.001 |

Abbreviations: CABG, coronary artery bypass graft; PCI, percutaneous coronary intervention.

[a] We divided the study populations into two groups according to whether there were presentations by doctors in the corresponding hospitals in the same year as the patient's admission year (Presentation group and Control group).

**Table 2. Odds ratios for in-hospital mortality associated with conference presentation.**

| | Odds ratio | 95% CI | C-statistics |
|---|---|---|---|
| Crude | 0.68 | 0.65 to 0.72 | 0.55 (0.54 to 0.55) |
| Model 1 [a] | 0.73 | 0.68 to 0.79 | 0.91 (0.91 to 0.92) |
| Model 2 [b] | 0.76 | 0.70 to 0.82 | 0.91 (0.91 to 0.92) |
| Model 3 [c] | 0.84 | 0.76 to 0.92 | - |
| Model 4 [d] | 1.00 | 0.92 to 1.09 | 0.94 (0.94 to 0.94) |

[a]: Model 1: adjusted for sex, age, Killip classification, smoking, ambulance use, hypertension, atrial fibrillation, old myocardial infarction, diabetes, renal disease, and chronic obstructive pulmonary disease

[b]: Model 2: adjusted for Model 1 plus admission year and the number of admissions

[c]: Model 3: multilevel analysis clustered by hospital codes and adjusted for the same variables as Model 1.

[d]: Model 4: adjusted for Model 1 plus the evidence-based practices through causal mediation analysis.

### Sensitivity analyses

Conference presentation was significantly associated with lower in-hospital mortality: 1) excluding university hospitals and their affiliated hospitals: OR = 0.76, 95% CIs: 0.70 to 0.82; 2) restricting patients to those who stayed more than two days: OR = 0.78, 95% CIs: 0.72 to 0.85; 3) restricting presentations to first author only: OR = 0.75, 95% CIs: 0.70 to 0.82; 4) excluding 9,802 patients without the Barthel index, then adjusting for the Barthel index: OR = 0.78, 95% CIs: 0.72 to 0.85; 5) excluding 871 patients readmitted during the observation period: OR = 0.76, 95% CIs: 0.71 to 0.80.

## Discussion

In our analysis using an administrative database that included data from 384 acute care hospitals, a larger number of conference presentations was significantly associated with lower in-hospital mortality in all multivariate logistic regression models used, and this association was also confirmed in sensitivity analyses. In addition, more patients admitted in the Presentation group tended to receive optimal medical therapies, according to treatment guidelines and/or evidence-based practice, compared with patients in the Control group.

Among the strengths of our study is its inclusion of the body of evidence in the cardiovascular field and its treatment of Killip classification as a confounding factor. The cardiovascular department has accumulated a body of evidence based on many randomized clinical trials [9], and greater adherence to treatment guidelines may bring about the lower mortality shown in our results. In our analysis, the c-statistic for Model 2 was 0.91 (95% CI: 0.91–0.92), meaning good discrimination, mainly because the important prognostic factors for AMI, including Killip classification, were included. Actually, several researchers have shown Killip classification to be a good prognostic factor [17, 18]. Similarly, conference presentations may be associated with lower in-hospital mortality in other departments, if those departments acted according to evidence-based practices and established clinical guidelines.

The study described here revealed an association between the number of conference presentations and patient mortality in cases of AMI. Past reports have shown that institutional or operator factors may influence the mortality of AMI [19–21], partially because conference presentations might be related to these associations. In multiple acute care hospitals, the presence of more board-certified cardiologists and lower institutional volume was associated with a lower risk of in-hospital mortality among patients with cardiovascular disease [20, 21]. On the other hand, the annual operator volume was not related to in-hospital mortality [20]. Once the JCS has certified a doctor as a cardiologist in Japan, these doctors must make presentations at JCS conferences to become cardiologists. Similarly, The Accreditation Council for Graduate Medical Education (ACGME) Common Program mandates faculty scholarly activities with an active research component and supports residents in their scholarly activities [11], while the Japanese Society of Internal Medicine imposes on residents the requirement of performing at least two scholarly activities in order to become internal medicine specialists. Plausibly, conference presentations may be an important mechanism to ensure the quality of medical care and an associated lower risk of in-hospital mortality.

Several studies have shown that scholarly activities by hospital physicians may lead to lower mortality, greater adherence to treatment guidelines and/or an increase in evidence-based practice [4, 7, 8]. Jha et al. studied the survival of AMI patients in hospitals conducting clinical trials and found that the survival rate for participants was higher than the survival rate for non-participants [8]. Similarly, Majumdar et al. reported that patients treated in hospitals whose physicians participated in trials showed significantly lower mortality than those treated in non-participating hospitals [7]. In this previous literature, a composite guideline adherence

score was used and the uptake of nine recommendations from the guidelines was assessed. It was found that the adherence proportion tended to increase significantly with the participation level. In our analyses, the patients in the Presentation group were consistently prescribed the five evidence-based drugs in more cases than those admitted to hospitals with no annual conference presentations. The difference in mortality in our results may be explained by these prescriptions.

There are several limitations to the study. First, "conference presentation" was defined as a presentation only at the annual scientific meetings of the JCS, which is recognized as one of the major medical societies in Japan in view of the importance of cardiovascular diseases. (As of June 2021, the JCS has 26,970 regular members, 4,570 associate members, and 15,098 cardiovascular specialists.) We have chosen the JCS conference based on the current healthcare performance, but it is fully recognized that similar effects could be achieved with another conference presentation, such as those held by the Japanese Society of Internal Medicine or the Japanese Association of Cardiovascular Intervention and Therapeutics. Second, unmeasured confounding hospital and personal factors exist, such as organizational culture, the cardiologist's motivation, and NYHA classification. Future observational studies with comprehensive measurements of these confounding factors would enhance our understanding of their effects on patient mortality. Thus, our cohort study might feasibly provide evidence that can be considered somewhat credible.

## Conclusion

We concluded that conference presentations were associated with lower in-hospital mortality, and patients tend to benefit from more evidence-based practice in hospitals whose physicians make conference presentations.

## Acknowledgments

The authors wish to thank all the staff members and all the participants in the hospitals participating in the QIP project.

## Author Contributions

**Conceptualization:** Daisuke Takada, Yuki Kataoka, Yuichi Imanaka.

**Investigation:** Daisuke Takada.

**Methodology:** Daisuke Takada.

**Writing – original draft:** Daisuke Takada.

**Writing – review & editing:** Yuki Kataoka, Tetsuji Morishita, Noriko Sasaki, Susumu Kunisawa, Yuichi Imanaka.

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
