## [Decision Letter · Decision Letter 0]

22 Oct 2024

PONE-D-24-39413The Relation between Conference Presentations and In-hospital Mortality in Patients Admitted with Acute Myocardial Infarction: A Retrospective Analysis Using a Japanese Administrative DatabasePLOS ONE

Dear Dr. Imanaka,

Thank you for submitting your manuscript to PLOS ONE. After careful consideration, we feel that it has merit but does not fully meet PLOS ONE’s publication criteria as it currently stands. Therefore, we invite you to submit a revised version of the manuscript that addresses the points raised during the review process.

We look forward to receiving your revised manuscript.

Kind regards,

Shukri AlSaif

Academic Editor

PLOS ONE

Journal Requirements:

'This study was funded by Ministry of Health, Labour and WelfareHealth and Labour Sciences Research Grants/Grant

(JPMH21IA1005, JPMH22AA2003) and Japan Society for the Promotion of Science (JP23H00448).'

Please state what role the funders took in the study.  If the funders had no role, please state: ''The funders had no role in study design, data collection and analysis, decision to publish, or preparation of the manuscript.'' 

Reviewers' comments:

Reviewer's Responses to Questions

**Comments to the Author**

1. Is the manuscript technically sound, and do the data support the conclusions?

Reviewer #1: Yes

2. Has the statistical analysis been performed appropriately and rigorously? 

Reviewer #1: Yes

3. Have the authors made all data underlying the findings in their manuscript fully available?

Reviewer #1: Yes

4. Is the manuscript presented in an intelligible fashion and written in standard English?

Reviewer #1: Yes

5. Review Comments to the Author

Reviewer #1: In this study, the authors demonstrated the association between conference presentations and in-hospital mortality in patients with myocardial infarction using a Japanese administrative database. While the research topic is of significant interest, there are several points that should be improved for better understanding. Please see the following comments to the authors.

Major comments

Abstract

1. The font size in the abstract is inconsistent with the rest of the manuscript. Please ensure that the font size is uniform throughout the document.

2. The hypothesis of this study is not explicitly stated in the abstract. Clearly stating the study hypothesis will help readers understand the rationale behind the research.

Methods

1. The section titled "Intervention" may not be the most appropriate terminology in this context. The term "Exposure" might be more fitting, considering the nature of the study. Please revise accordingly.

2. The Ethical Consideration and Data Availability sections contain overlapping information with the previous sections. To improve clarity and avoid redundancy, please condense or reorganize these sections.

3. It is unclear what specific information was collected from the database. A more detailed description of the type and scope of the collected data is needed for transparency.

4. The authors should consider including additional variables for adjustment, particularly ADL prior to hospitalization, NYHA classification, and the presence of pulmonary disease. These are important confounding factors that could impact the study's findings.

5. The authors used multivariable logistic regression analysis; however, given that both individual-level and hospital-level factors are being examined, a multi-level analysis may be more suitable. Please clarify whether a multi-level analysis was conducted. If not, the authors should provide a rationale for this choice.

6. Please consider adding hospital variables such as whether the hospitals are university-affiliated or their geographical location.

7. The handling of readmissions in the analysis is unclear. Please provide details on how readmission data was managed and its potential impact on the study outcomes.

8. Please clarify if there is any relationship between the number of conference presentations and in-hospital mortality, as this could provide additional insights into the factors affecting patient outcomes.

Other comments

1. Proof-reading is required to address several grammatical and stylistic issues throughout the manuscript.

6. PLOS authors have the option to publish the peer review history of their article (what does this mean?). If published, this will include your full peer review and any attached files.

Reviewer #1: No

---

## [Author Response · Author response to Decision Letter 0]

5 Nov 2024

Response to the Reviewers

Major comments

Abstract

1. The font size in the abstract is inconsistent with the rest of the manuscript. Please ensure that the font size is uniform throughout the document.

Thank you for pointing this out, and sorry for the inconsistent font size. We have revised font sizes to ensure consistency throughout the document. 

2. The hypothesis of this study is not explicitly stated in the abstract. Clearly stating the study hypothesis will help readers understand the rationale behind the research.

Thank you for your suggestion. We have clearly stated our hypothesis in the abstract and introduction. 

Abstract(P3 L31-35): ”Background and aims: Clinicians’ research activities reportedly improve their healthcare performance. Presenting research at conferences may be related to improved patient care outcomes; however, few studies have investigated this relationship. Thus, we examined the association between presenting at conferences and the mortality of patients hospitalized for acute myocardial infarction..”

Introduction(P7 L90-95): “Our hypothesis is that clinicians’ conference presentations might be associated with improved patient outcomes through increased evidence-based practice. Therefore, we examined the association between clinicians’ conference presentations and mortality in acute myocardial infarction (AMI) cases after adjusting for potentially confounding patient characteristics. We further confirmed that the proportions of prescribed medical therapies were optimal according to treatment guidelines for evidence-based practice.”

Methods

1. The section titled "Intervention" may not be the most appropriate terminology in this context. The term "Exposure" might be more fitting, considering the nature of the study. Please revise accordingly.

Thank you for your suggestion. In our manuscript, we have changed the term to “Exposure.” (P9 L119)

2. The Ethical Consideration and Data Availability sections contain overlapping information with the previous sections. To improve clarity and avoid redundancy, please condense or reorganize these sections.

Thank you for pointing this out, and we apologize for the overlapping information. We have revised the manuscript. (P11 L159 - P12 L164 )

“Ethical consideration and Data availability

The Ethics Committee, Graduate School of Medicine, Kyoto University, approved this study (approval number: R0135). This study was conducted in accordance with the Ethical Guidelines for Medical and Health Research Involving Human Subjects of the Ministry of Health, Labour and Welfare, Japan. These guidelines require the protection of patient anonymity. The data were anonymized, and the requirement for informed consent was waived.”

3. It is unclear what specific information was collected from the database. A more detailed description of the type and scope of the collected data is needed for transparency.

Thank you for your suggestion. We have added more details about the data we collected. (P8 L104 - 111)

“The DPC data comprises a Form 1, an EF-file, and other files. Form 1 contains important clinical summaries, including date of birth, sex, admission dates, discharge outcome, diagnosis, surgical procedure, and various severity scores such as the Killip classification upon admission and other clinical information. The International Classification of Diseases, 10th Revision (ICD-10) codes were used for diagnosis in the DPC database. The EF-file contains information on medical resource utilization during hospitalization, including medications administered and other medical procedures.”

4. The authors should consider including additional variables for adjustment, particularly ADL prior to hospitalization, NYHA classification, and the presence of pulmonary disease. These are important confounding factors that could impact the study's findings.

Thank you for your suggestion about the additional variables for adjustment. We had chosen the confounding factors that refer to past reports and clinical experience as physicians (some authors), but we considered the additional variables again and added several variables as outlined below. 

・The presence of pulmonary disease: We added chronic obstructive pulmonary disease as a key pulmonary comorbidity based on the Charlson Comorbidity Index (1-3).

Method(P10 L136-139): “Model 1 was adjusted for sex, age, the Killip classification, smoking, ambulance use, hypertension, atrial fibrillation, old myocardial infarction, diabetes, renal disease and chronic obstructive pulmonary disease (personal factors)”

・Patient ADL data were often missing from the database, so we performed the analysis using the ADL variable in the sensitivity analysis. 

Method(P11 L148-150): “4) additionally adjusting for the Barthel index when the data did not contain any missing measurements ”

Result(P16 L218-219): “4) excluding 9,802 patients without the Barthel index, then adjusting for the Barthel index: OR=0.78, 95% CIs: 0.72 to 0.85.”

・NYHA classifications would be included in the database if the patients were diagnosed with heart failure and the most resources were spent on heart failure treatment during hospitalization. However, because heart failure was not the primary diagnosis in this analysis, the NYHA classifications were rarely included in the research data; hence, we cannot adjust the NYHA classification variable in the revised manuscript. We have added this as a limitation in the revised manuscript, as outlined below.

Discussion(P20 L280 - P21 L284):“Second, unmeasured confounding hospital and personal factors exist, such as organizational culture, the cardiologist’s motivation, and NYHA classification. Future observational studies with comprehensive measurements of these confounding factors would enhance our understanding of their effects on patient mortality.” 

5. The authors used multivariable logistic regression analysis; however, given that both individual-level and hospital-level factors are being examined, a multi-level analysis may be more suitable. Please clarify whether a multi-level analysis was conducted. If not, the authors should provide a rationale for this choice.

Thank you for your suggestion. We conducted a multilevel analysis and provide the results (Model 3) in our manuscript. 

Method (P10 L141 - 142): “Model 3 was a multilevel analysis clustered by hospital codes and adjusted for the same variables as Model 1.”

Result (P15 L199): “OR=0.84, 95% CIs: 0.76 to 0.92 in Model 3;”

Table 2 (P15 L202 – P16 L211): 

 Odds ratio 95% CI C-statistics 

Crude 0.68 0.65 to 0.72 0.55 (0.54 to 0.55)

Model 1 a 0.73 0.68 to 0.79 0.91 (0.91 to 0.92)

Model 2 b 0.76 0.70 to 0.82 0.91 (0.91 to 0.92)

Model 3 c 0.84 0.76 to 0.92 -

Model 4 d 1.00 0.92 to 1.09 0.94 (0.94 to 0.94)

a: Model 1: adjusted for sex, age, Killip classification, smoking, ambulance use, hypertension, atrial fibrillation, old myocardial infarction, diabetes, renal disease, and chronic obstructive pulmonary disease

b: Model 2: adjusted for Model 1 plus admission year and the number of patients

c: Model 3: multilevel analysis clustered by hospital codes adjusted for the same variables as Model 1

d: Model 4: adjusted for Model 1 plus the evidence-based practices as causal mediation analysis

6. Please consider adding hospital variables such as whether the hospitals are university-affiliated or their geographical location.

Thank you for your suggestion. As you say, clinicians from university-affiliated hospitals may affect the results of this research because we think that they may have given presentations about basic research. Therefore, we excluded the university-affiliated hospitals in the sensitivity analysis and confirmed that the results were similar to that of the main analysis. 

Result(P16 L215): “1) excluding university hospitals and their affiliated hospitals: OR=0.76, 95% CIs: 0.70 to 0.82;)”

In contrast, it was difficult to adjust our models for geographical location. We consider the results of Model 3 (a multilevel analysis clustered by hospital codes) to show the results adjusted by alternative hospital variables to some extent.

Table 2 (P15 L202 – P16 L211): 

 Odds ratio 95% CI C-statistics 

Crude 0.68 0.65 to 0.72 0.55 (0.54 to 0.55)

Model 1 a 0.73 0.68 to 0.79 0.91 (0.91 to 0.92)

Model 2 b 0.76 0.70 to 0.82 0.91 (0.91 to 0.92)

Model 3 c 0.84 0.76 to 0.92 -

Model 4 d 1.00 0.92 to 1.09 0.94 (0.94 to 0.94)

a: Model 1: adjusted for sex, age, Killip classification, smoking, ambulance use, hypertension, atrial fibrillation, old myocardial infarction, diabetes, renal disease, and chronic obstructive pulmonary disease

b: Model 2: adjusted for Model 1 plus admission year and the number of patients

c: Model 3: multilevel analysis clustered by hospital codes adjusted for the same variables as Model 1

d: Model 4: adjusted for Model 1 plus the evidence-based practices as causal mediation analysis

7. The handling of readmissions in the analysis is unclear. Please provide details on how readmission data was managed and its potential impact on the study outcomes.

Thank you for your suggestion. If you mean the readmission data leading to the potential impact denoted in the intra-individual correlation, we considered that the results of the multilevel analysis clustered by hospital codes (Model 3) also showed the results adjusted by alternative hospital variables to some extent. 

Moreover, when we limited our analysis to only the first hospitalization during the observation period, the odds ratio was 0.76 (0.71–0.80), which was similar to the results of Model 2. We added this result of the sensitivity analysis to the manuscript.

Method(P11 L150-151): 5) excluding patients who were readmitted during the observation period .

Result(P16 L219 – P17 L220): 5) excluding 871 patients readmitted during the observation period: OR=0.76, 95% CIs: 0.71 to 0.80.

8. Please clarify if there is any relationship between the number of conference presentations and in-hospital mortality, as this could provide additional insights into the factors affecting patient outcomes.

To provide additional insights into the causal relationship, we have now performed two further analyses: Causal Mediation Analysis and Generalized Additive Modeling (spline). 

First, Causal Mediation Analysis (Model 4 in our revised manuscript) showed that clinicians’ conference presentations affected patient outcomes through evidence-based practice, evidenced by the odds ratios being significantly lower in Model 1 but not in Model 4.

Method(P10 L142-143): Model 4 was adjusted for Model 1 plus the evidence-based practices through causal mediation analysis.

Result(P15 L199 – L200): OR=0.96, 95% CIs: 0.88 to 1.05 in Model 4

Table 2 (P15 L202 – P16 L211): 

 Odds ratio 95% CI C-statistics 

Crude 0.68 0.65 to 0.72 0.55 (0.54 to 0.55)

Model 1 a 0.73 0.68 to 0.79 0.91 (0.91 to 0.92)

Model 2 b 0.76 0.70 to 0.82 0.91 (0.91 to 0.92)

Model 3 c 0.84 0.76 to 0.92 -

Model 4 d 1.00 0.92 to 1.09 0.94 (0.94 to 0.94)

a: Model 1: adjusted for sex, age, Killip classification, smoking, ambulance use, hypertension, atrial fibrillation, old myocardial infarction, diabetes, renal disease, and chronic obstructive pulmonary disease

b: Model 2: adjusted for Model 1 plus admission year and the number of patients

c: Model 3: multilevel analysis clustered by hospital codes adjusted for the same variables as Model 1

d: Model 4: adjusted for Model 1 plus the evidence-based practices as causal mediation analysis

Second, Generalized Additive Modeling (spline) revealed the volume-outcome relationship between the number of presentations per year (horizontal axis value) and patient mortality (below).

We can graphically depict the causal relationship, but it is difficult to interpret the value of the vertical axis clinically, so we have not included this figure in the revised manuscript.

If the editor and reviewers request these outcomes be included in the revised manuscript, we will include them.

Other comments

1. Proof-reading is required to address several grammatical and stylistic issues throughout the manuscript.

 We thank Cambridge English Correction Service (https://cambridge-correction.com) for English language editing, and we have rechecked English grammar and the logical flow of information in the revised manuscript.

Reference

1. Charlson ME, Pompei P, Ales KL, MacKenzie CR. A new method of classifying prognostic comorbidity in longitudinal studies: development and validation. J Chronic Dis. 1987;40(5):373-83.

2. Quan H, Li B, Couris CM, Fushimi K, Graham P, Hider P, et al. Updating and validating the Charlson comorbidity index and score for risk adjustment in hospital discharge abstracts using data from 6 countries. American journal of epidemiology. 2011;173(6):676-82.

3. Hautamäki M, Lyytikäinen LP, Mahdiani S, Eskola M, Lehtimäki T, Nikus K, et al. The association between charlson comorbidity index and mortality in acute coronary syndrome - the MADDEC study. Scand Cardiovasc J. 2020;54(3):146-52.

---

## [Decision Letter · Decision Letter 1]

22 Nov 2024

The Relation between Conference Presentations and In-hospital Mortality in Patients Admitted with Acute Myocardial Infarction: A Retrospective Analysis Using a Japanese Administrative Database

PONE-D-24-39413R1

Dear Dr. Imanaka,

We’re pleased to inform you that your manuscript has been judged scientifically suitable for publication and will be formally accepted for publication once it meets all outstanding technical requirements.

Kind regards,

Shukri AlSaif

Academic Editor

PLOS ONE

Additional Editor Comments (optional):

Reviewers' comments:

Reviewer's Responses to Questions

**Comments to the Author**

1. If the authors have adequately addressed your comments raised in a previous round of review and you feel that this manuscript is now acceptable for publication, you may indicate that here to bypass the “Comments to the Author” section, enter your conflict of interest statement in the “Confidential to Editor” section, and submit your "Accept" recommendation.

Reviewer #1: All comments have been addressed

2. Is the manuscript technically sound, and do the data support the conclusions?

Reviewer #1: Yes

3. Has the statistical analysis been performed appropriately and rigorously? 

Reviewer #1: Yes

4. Have the authors made all data underlying the findings in their manuscript fully available?

Reviewer #1: Yes

5. Is the manuscript presented in an intelligible fashion and written in standard English?

Reviewer #1: Yes

6. Review Comments to the Author

Reviewer #1: The authors have adequately revised the manuscript according to the reviewers' recommendations. This version of the manuscript seems suitable for publication.

7. PLOS authors have the option to publish the peer review history of their article (what does this mean?). If published, this will include your full peer review and any attached files.

Reviewer #1: No

---

## [Editor Report · Acceptance letter]

28 Nov 2024

PONE-D-24-39413R1 

PLOS ONE

Dear Dr. Imanaka, 

I'm pleased to inform you that your manuscript has been deemed suitable for publication in PLOS ONE. Congratulations! Your manuscript is now being handed over to our production team.

Kind regards, 

on behalf of

Dr. Shukri AlSaif 

Academic Editor

PLOS ONE